# Making the Shoe Fit: Architectures, Initializations, and Tuning for Learning with Privacy

## Abstract

Because learning sometimes involves sensitive data, standard machine-learning algorithms have been extended to offer strong privacy guarantees for training data. However, in practice, this has been mostly an afterthought, with privacy-preserving models obtained by re-running training with a different optimizer, but using the same model architecture that performed well in a non-privacy-preserving setting. This approach leads to less than ideal privacy/utility tradeoffs, as we show here. Instead, we propose that model architectures and initializations are chosen and hyperparameter tuning is performed, *ab initio*, explicitly for privacy-preserving training. Using this paradigm, we achieve new state-of-the-art accuracy on MNIST, FashionMNIST, and CIFAR10 without any modification of the fundamental learning procedures or differential-privacy analysis.

## 1 Introduction

Machine learning (ML) can be usefully applied to the analysis of sensitive data, e.g., in the domain of healthcare (Kononenko, 2001). However, ML models may unintentionally reveal sensitive aspects of their training data, e.g., due to overfitting (Shokri et al., 2017; Song & Shmatikov, 2019). To counter this, ML techniques that offer strong privacy guarantees have been developed. Notably, the differentially private stochastic gradient descent, or DP-SGD, of Abadi et al. (2016) is an easy-to-use, generally-applicable modification of stochastic gradient descent. In addition to its rigorous privacy guarantees, it has been empirically shown to stop the leaking of secrets (Carlini et al., 2019).

To strictly bound the impact of any training example, DP-SGD makes two changes to every gradient step: first, each example's gradient contribution is limited to a fixed bound (in practice, by clipping all per-example gradients to a maximum $\ell_2$ norm); second, random (Gaussian) noise of the scale of the clipping norm is added to each batch's combined gradient, before it is backpropagated to update model parameters. Together, these changes create a new, artificial noise floor at each step of gradient descent, such that the unique signal of any individual example is below this new noise floor; this allows differential privacy to be guaranteed for all training examples (Dwork & Roth, 2014).

Training using DP-SGD is eminently practical and in addition to privacy offers advantages such as strong generalization and the promise of reusable holdouts (Google, 2019; Dwork et al., 2015). Unfortunately, its advantages have not been without cost: empirically, the test accuracy of differentially private ML is consistently lower than that of non-private learning (e.g., see Papernot et al. (2018)). Such accuracy loss may sometimes be inevitable: for example, the task may involve heavy-tailed distributions and adding noise will definitely hinder visibility of examples in the tails (Feldman, 2019; Bagdasaryan & Shmatikov, 2019). However, this does not explain the accuracy loss of differentially private learning on standard benchmark tasks that are known to be relatively simple: MNIST (Yann et al., 1998), FashionMNIST (Xiao et al., 2017), CIFAR10 (Krizhevsky et al., 2009), etc.

This paper presents several new results for privacy-preserving learning that improve the state-of-the-art in terms of both privacy and accuracy. Significantly, these new results stem from a single, simple observation: differentially-private learning with DP-SGD is different enough that all aspects of learning—model architecture, parameter initialization, and optimization strategy, as well as hyperparameter tuning—must be reconsidered. To achieve the best privacy/accuracy tradeoffs, we must tune our learning strategies to the specifics of privacy-preserving learning; i.e., we must "learn to learn" with privacy. Conversely, we concretely demonstrate how the architecture, initialization,

and optimization strategy that gives the best accuracy for non-private learning can be a poor fit for learning with privacy. Instead, by revisiting our choices, we can reduce the information loss induced by clipping, limit the impact of added noise, and improve the utility of each gradient step when learning with privacy. Our contributions facilitate DP-SGD learning as follows:

- We show how simple architecture changes, such as the use of `tanh` instead of ReLU activations, can improve a model's private-learning suitability and achievable privacy/accuracy tradeoffs, by eliminating the negative effects of clipping and noising large gradients.
- We explain how high-capacity models can be disadvantageous, as well as the advantages of models with a final, fully-connected layer that can be independently fine tuned, and how both help address the curse of dimensionality and high-dimensional noise.
- We demonstrate the importance of finding good initializations, and show how this can be done with privacy using either transfer learning or weight scaling (Raghu et al., 2019).
- We show that better tradeoffs and increased wall-clock learning speeds can be achieved by tuning hyperparameters and choosing optimizers directly for DP-SGD learning.

By applying the above, we advance the state of the art for MNIST, FashionMNIST, and CIFAR10, significantly improving upon the privacy/accuracy tradoffs from prior work. On MNIST, we achieve 98.1% test accuracy for a privacy guarantee of $(\varepsilon, \delta) = (2.93, 10^{-5})$, whereas the previous state-of-the-art reported in the TensorFlow Privacy library (Google, 2019) was 96.6%. On CIFAR10, we achieve 72% test accuracy at $(\varepsilon, \delta) = (2.1, 10^{-5})$ in a setup for which to the best of our knowledge the previous state-of-the-art was achieved by Abadi et al. (2016) at 67% accuracy.

## 2 TRAINING-DATA MEMORIZATION, DIFFERENTIAL PRIVACY, AND DP-SGD

Machine-learning models will easily memorize whatever sensitive, personal, or private data that was used in their training, and models may in practice disclose this data—as demonstrated by the attacks of Shokri et al. (2017), Song & Shmatikov (2019), and Carlini et al. (2019).

For reasoning about the privacy guarantees of algorithms such as training by stochastic gradient descent, differential privacy has become the established gold standard (Dwork & Roth, 2014). Informally, an algorithm can be differentially private if it will always produce effectively the same output (in a mathematically precise sense), when applied to two input datasets that differ by only one record. Formally, a learning algorithm $A$ that trains models from the set $S$ is $(\varepsilon, \delta)$-differentially-private, if the following holds for all training datasets $d$ and $d'$ that differ by exactly one record:

$$Pr[A(d) \in S] \leq e^{\varepsilon} Pr[A(d') \in S] + \delta \tag{1}$$

Here, $\varepsilon$ gives the formal privacy guarantee, by placing a strong upper bound on any privacy loss, even in the worst possible case. A lower $\varepsilon$ indicates a stronger privacy guarantee or a tighter upper bound (Erlingsson et al., 2019). The factor $\delta$ allows for some probability that the property may not hold (in practice, this $\delta$ is required to be very small, e.g., in inverse proportion to the dataset size).

A very attractive property of differential-privacy guarantees is that they hold true for all attackers—whatever they are probing and whatever their prior knowledge—and that they remain true under various forms of composition. In particular, the output of a differentially-private algorithm can be arbitrarily post processed, without any weakening of the guarantees. Also, if sensitive training data contains multiple examples from the same person (or, more generally, the same sensitive group), $\varepsilon$-differentially-private training on this data will result in model with a $k\varepsilon$-differential-privacy guarantee for each person, as long as at most $k$ training-data records are present per person.

Abadi et al. (2016) introduced DP-SGD as a method for training deep neural networks with differential-privacy guarantees that was able to achieve better privacy and utility than previous efforts (Chaudhuri et al., 2011; Song et al., 2013; Bassily et al., 2014). DP-SGD bounds the sensitivity of the learning process to each individual training example by computing per-example gradients $\{g_i\}_{i \in 0..n-1}$ with respect to the loss, for the $n$ model parameters $\{\theta_i\}_{i \in 0..n-1}$, and clipping each per-example gradient to a maximum fixed $\ell_2$ norm $C$. Subsequently, to the average of these per-example gradients, DP-SGD adds (Gaussian) noise that whose standard deviation $\sigma$ is proportional to this sensitivity. In this work, we use the canonical implementation of DP-SGD and its associated analysis that has been made available through the TensorFlow Privacy library (Google, 2019).

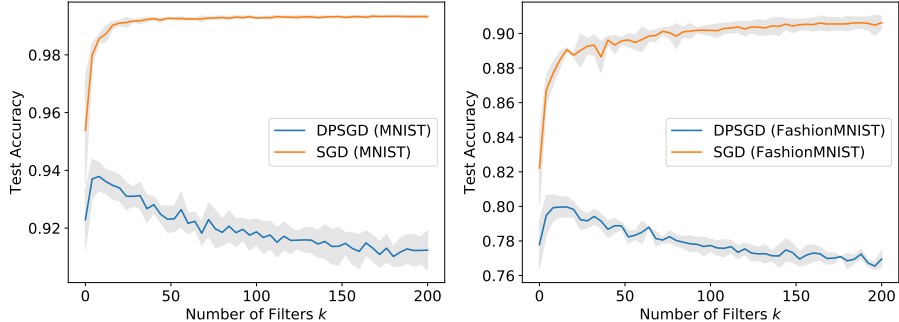

Figure 1: Test accuracy as a function of the number of filters $k$ in the convolutional architecture of Table 1; when training with vanilla SGD and DPSGD. Each point corresponds to multiple training runs on MNIST (left) or FashionMNIST (right). For both datasets, adding filters always improves non-private learning, whereas after an early point they are not beneficial to learning with privacy.

## 3    MODEL ARCHITECTURES BETTER SUITED TO LEARNING WITH PRIVACY

We show here that learning with differential privacy imposes additional constraints that need to be taken into account when designing neural network architectures. They help us control the sensitivity of learning to training examples before the clipping operation is performed in DP-SGD, thus reducing the potential negative impact of clipping on the estimated gradient direction.

### 3.1    MODEL CAPACITY

The success of neural networks is in part explained by their ability to scale to complex tasks through an increase in model capacity. ResNets are an illustrative recent examples (He et al., 2016). Here, we explain how additional capacity may *not* be beneficial when learning with privacy. One of the major challenges in training models with differential privacy is the *curse of dimensionality* (Bassily et al., 2014). The accuracy of privately trained models typically degrades with the increase in the number of dimensions. Unfortunately, strong lower bounds suggest that this dependence on dimensionality is *necessary* (Bassily et al., 2014).

Consider the convolutional architecture described to the right. With all other architectural details being fixed, we can control the model's capacity by varying the number of filters $k$ in its two convolutional layers. While the relationship between generalization performance and the number of parameters is not always monotonic (Neyshabur et al., 2017), we leave as future work a study of how different measures of capacity can inform the design of model architectures for private learning. We report the model's accuracy when trained with SGD

Table 1: MNIST and FashionMNIST model architecture (33,000 parameters for $k = 31$).

| Layer | Parameters |
|---|---|
| Convolution | $k$ filters of 8x8, strides 2 |
| Max-Pooling | 2x2 |
| Convolution | $k$ filters of 4x4, strides 2 |
| Max-Pooling | 2x2 |
| Fully connected | 32 units |
| Softmax | 10 units |

and DP-SGD in Figure 1, both on MNIST (left) and FashionMNIST (right). The test accuracy of models trained without privacy monotonically increases with the number of filters in their convolutional layers. Instead, we observe an inflection point at about 15 filters for which models trained with privacy achieve their highest test accuracy. Afterwards, the model's generalization suffers as more filters are added.

There are two competing explanations of this behavior, both compatible with the lower bound stated in Bassily et al. (2014). First, recall that DP-SGD performs a clipping operation on each per-example gradient before the average gradients is used to update model parameters; i.e., each gradient is subject to the following transformation

$$g_i \leftarrow g_i \cdot \min \left( 1, \frac{C}{\sqrt{\sum_{i=0}^{n-1} g_i^2}} \right) \tag{2}$$

where $g_i$ is the gradient corresponding to model parameter $i$. For a fixed clipping norm $C$ (corresponding to a certain, fixed privacy guarantee), the quantity $\frac{C}{\sqrt{\sum_{i=0}^{n-1} g_i^2}}$ by which individual parameters are multiplied decreases as the number $n$ of parameters in a model increases. That is, the more

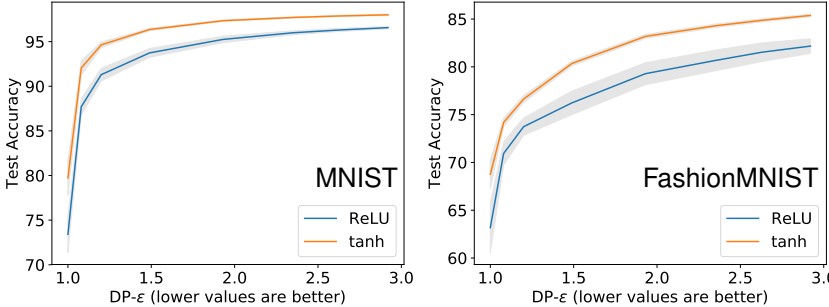

Figure 2: Test accuracy as a function of the privacy loss when training a pair of models with DP-SGD. The only difference between the two models is the activation function for their hidden layer: ReLU or tanh. All other elements of the architecture (number, type, and dimension of layers) and the training algorithm (optimizer, learning rate, number of microbatches, clipping norm, and noise multiplier) are identical. Results are averaged over 10 runs for each curve.

parameters we have, the more likely DP-SGD is to clip the gradient (or signal) at each parameter. This can explain the presence of an inflection point in Figure 1, after which learning with privacy becomes increasingly difficult as capacity is increased. Second, as the number of parameters (i.e., $g_i$'s) increases, the norm of the noise vector that DP-SGD must add to the gradient average to ensure privacy also increases. This noise norm increases as $\sqrt{\#\text{parameters}}$, and introduces another source of accuracy degradation with an increased number of parameters.

Our observations may seem to contradict some of the findings in Abadi et al. (2016). However, their limited experimental setup could offer few general lessons. First, they reduced data dimensionality using PCA to have inputs of only 60 dimensions; second, they explored only a model architectures using a single layer perceptron with between 200 and 2,000 units. Instead, our experiments involve a realistic setting where the full input is passed to a convolutional neural network with a total of 3 hidden layers and over 26,000 parameters.

### 3.2 ACTIVATION FUNCTIONS

When training a model with differential privacy, gradients computed during SGD are clipped (recall Equation 2) to control the sensitivity of learning to training examples. If these gradients take large values, some of the signal will be discarded as gradients are being clipped. One way to reduce the magnitude (or at least control it), is to prevent the model's activations from exploding. However, a common choice of activation function in modern deep neural networks is the ReLU and, unlike other activations functions, ReLUs are unbounded.

Here, we thus test the hypothesis that replacing ReLUs with a bounded activation function prevents activations from exploding and thus keeps the magnitude of gradients to a more reasonable value. This in turn implies that the clipping operation applied by DP-SGD will discard less signal from gradient updates—eventually resulting in higher performance at test time.

On MNIST and FashionMNIST, we train two models based off the architecture of Table 1: the first model uses ReLU whereas the second model uses tanh[1] as the activation for its hidden layers, with other architectural elements kept identical. In our experiments, we later fine-tuned those architectural aspects (i.e., model capacity, choice of optimizer, etc.) separately for each activation function, to avoid favoring any one choice. In all cases, tanh was an improvement, as summarized in our conclusions (Section 6).

Figure 2 visualizes the privacy-utility Pareto curve (Avent et al., 2019) of the two models trained with DP-SGD. Rather than plotting the test accuracy as a function of the number of steps, we plot it as a function of the privacy loss $\varepsilon$ (but the privacy loss is a monotonically increasing function of the number of steps). On MNIST, the test accuracy of the tanh model is 98.0% compared to 96.6% for the ReLU model with an identical privacy loss of $\varepsilon = 2.93$. For comparison, baseline tanh and ReLU models trained without privacy both achieve a test accuracy of 99.0%. Similarly, on FashionMNIST, the tanh model trained with DP-SGD achieves 85.5% test accuracy compared to 81.9% with ReLUs. The baselines on FashionMNIST are 89.3% for tanh and 89.4% with ReLUs.

---

[1]We obtained results similar to the tanh with a sigmoid and a learning rate increased by a factor of 2 to 8. This is explained by the fact that the tanh is a rescaled sigmoid $\phi$: $\tanh(x) = 2\phi(x) - 1$.

To explain why a simple change of activation functions has a large impact on the model's accuracy, we conjecture that the bounded nature of the tanh prevents activations from exploding during training. We thus monitored the $\ell_2$ norm of the first layer's activations for our MNIST model while it is being trained in three scenarios: (a) without privacy using vanilla SGD and ReLU activations, (b) with ReLU activations and DP-SGD, and (c) with tanh activations and DP-SGD. The evolution of activation norms on test data is visualized in Figure 3. As conjectured, the activations of our ReLU network explode by a factor of 3 when training with privacy when compared to without privacy. Switching to tanh activations brings down the norms of activations back to levels comparable with the activations of our non-private ReLU network.

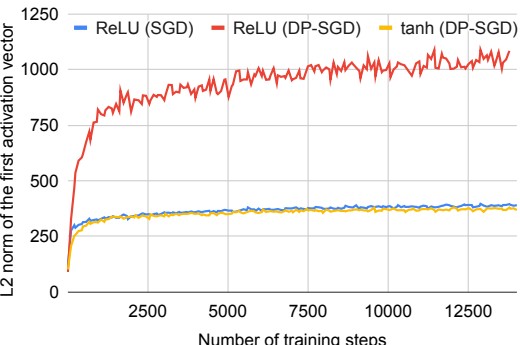

Figure 3: $\ell_2$ norm of the first conv activations.

# 4 INITIALIZATIONS FOR LEARNING WITH DIFFERENTIAL PRIVACY

Because each gradient step expends some privacy budget, good initialization of learning is important; here, we consider transfer learning (Pratt et al., 1991) and weight scaling (Raghu et al., 2019).

## 4.1 INITIALIZING FROM A PRE-TRAINED MODEL USING TRANSFER LEARNING

Transfer learning can improve the initialization used when learning with privacy, and allow better privacy/accuracy tradeoffs to be achieved.[2] For example, to reach reasonable accuracy ($> 80\%$) on CIFAR10, a convolutional neural network may necessarily include many convolutional layers comprising several hundred-thousand parameters. However, since convolutional layers for similar image-processing tasks are known to learn similar representations—at least in early layers—it may be possible to transfer most of these parameters from a public model, either as initializations or as frozen parameters, and subsequently train with DP-SGD. For CIFAR10, the natural choice for such transfer is a CIFAR100 model, and this has been previously explored by Abadi et al. (2016).

Taking the Abadi et al. (2016) transfer learning results for CIFAR10 as a baseline, we perform new experiments using much of the same setup and the model architecture of Table 2. As it is relatively simple, this model is a good candidate for differentially-private learning (although it reaches only $84.2\%$ accuracy on CIFAR10 when all its parameters are trained non-privately, whereas state-of-the-art models can have over 10% higher accuracy).

Table 2: CIFAR10 convolutional model architecture (in total, 2,395,434 parameters).

| | |
|---|---|
| Conv $\times$ 2 | 32 filters of 3x3, strides 1 |
| Max-Pooling | 2x2 |
| Conv $\times$ 2 | 64 filters of 3x3, strides 1 |
| Max-Pooling | 2x2 |
| Conv $\times$ 2 | 128 filters of 3x3, strides 1 |
| Fully connected | 1024 units |
| Softmax | 10 units |

We performed new transfer-learning experiments based on training this model on CIFAR100 data in three different ways: trained on a total of 5000 examples from 10 classes picked at random (**Min-rand-10**); trained on 25,000 examples from a random half of the CIFAR100 classes, grouped into 10 new, evenly-sized meta classes (**Half-rand-50**); trained on all examples and all 100 separate classes (**Max-100**). From each of these trained models, transfer learning was used to initialize a model to be trained on CIFAR10. In the subsequent CIFAR10 training, all but the last layer was frozen, which simplifies the learning task to that of logistic regression (but also reduces utility, with the best non-private accuracy reduced to 75% on CIFAR10).

Table 3 shows CIFAR10 privacy and accuracy resulting from fine-tuning of different transfer-learning models with DP-SGD. As shown in Table 4, the results improve on those of Abadi et al. (2016), even though they performed non-linear fine-tuning of two neural-network layers, and their underlying model was able to achieve higher non-private accuracy (86%).

---

[2]A different, formal take on how public models and data can facilitate learning with privacy is studied in (Bassily et al., 2018; Feldman et al., 2018).

| Type | Epoch 10 | Epoch 50 | Epoch 100 | Epoch 200 | Epoch 400 |
|---|---|---|---|---|---|
| **Min-rand-10** | 44.8% ± 4.6 | 49.6% ± 3.9 | 51.0% ± 3.9 | 52.8% ± 3.3 | 53.7% ± 3.5 |
| (81.0% ± 4.0) | 50% = best | 54.1% = best | 55.7% = best | 56.9% = best | 57.6% = best |
| **Half-rand-50** | 39.4% ± 2.9 | 51.4% ± 0.8 | 54.7% ± 1.5 | 56.8% ± 1.3 | 59.0% ± 0.9 |
| (62.1% ± 1.4) | 44.3% = best | 52.6% = best | 56.6% = best | 58.3% = best | 60.2% = best |
| **Max-100** | 57.0% ± 1.0 | 66.2% ± 0.6 | 68.4% ± 0.6 | 69.7% ± 0.6 | 71.0% ± 0.5 |
| (54.9% ± 0.7) | 59.1% = best | 67.2% = best | 69.5% = best | 70.6% = best | 72.1% = best |

Table 3: Accuracy of learning with privacy (average/best of 10 runs) compared to a non-private baseline of 75%. A CIFAR10 model is trained from a CIFAR100-transfer-learning initialization, with all-but-the-last layer frozen during training. The DP-SGD $\varepsilon$ upper bounds at $\delta = 10^{-5}$ are $\varepsilon_{10} = 0.32$, $\varepsilon_{50} = 0.73$, $\varepsilon_{100} = 1.04$, $\varepsilon_{200} = 1.48$, $\varepsilon_{400} = 2.12$ for the subscript-indicated epochs. The source model CIFAR100 accuracy (first column), is uncorrelated to the CIFAR10 accuracy.

In addition, the results show the benefits of model architectures whose final layer can be fine-tuned using logistic regression training, or other forms of convex optimization. Such training can be made possible by including a final fully-connected layer into a network; in additional experiments (not detailed here), the inclusion of such a layer did not harm the training of the original, source model from which transfer learning was done. Furthermore, the number of parameters in this layer did not seem to matter much: privacy/accuracy tradeoffs remained the same, even when the layer was grown by an order of magnitude, which is consistent with what is known about differentially-private convex optimization (Jain & Thakurta, 2014).

Table 4: CIFAR10 privacy and accuracy tradeoffs.

| This paper ($\varepsilon$, acc.) | Abadi et al. ($\varepsilon$, acc.) |
|---|---|
| (0.3, 59%) | − |
| (1.0, 70%) | − |
| (2.1, 72%) | (2.0, 67%) |
| − | (4.0, 70%) |
| − | (8.0, 73%) |

## 4.2 Initialization by Weight Scaling

Initialization by transfer learning is only applicable when a suitable public model exists whose weights can facilitate learning with privacy on sensitive data. But, such model may not exist, and DP-SGD learning may possibly benefit from other, non-standard means of initialization. We consider the *Mean Var* weight-scaling approach of Raghu et al. (2019) and initialize DP-SGD learning with Gaussian random parameter distributions whose layer-wise mean and variance are extracted from a seed model trained on the same sensitive input data. The weight-scaling approach does not directly transfer the parameters of an existing model; instead, just the layer-wise mean and variance are extracted, and those statistics are used to configure the Gaussian random distributions from which a second model with the same architecture is initialized.

In the context of learning with privacy, *Mean Var* weight scaling can improve model initialization by transfer from one differentially-private model to another. First, DP-SGD can be applied to train a model with standard random initialization. From this model, per-layer mean/variance statistics can be extracted to initialize a new model of the same architecture, sub-

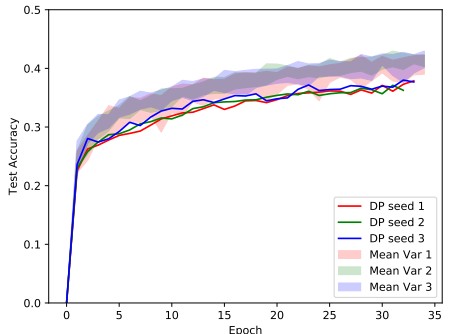

Figure 4: Colored lines show DP-SGD accuracy for three "seed" random initializations of a CIFAR-10 model. Colored bands show accuracy range of 30 DP-SGD models using *Mean Var* initialization based on per-layer parameter statistics in the corresponding seed model. In all models, the privacy $\varepsilon$ at each epoch is identical; however, *Mean Var* initialization substantially improves the privacy/accuracy tradeoff.

sequently trained with strong privacy guarantees. (This extraction can be done privately, although the privacy risk of summary statistics that drive random initialization should be vanishing. Following Bassily et al. (2018); Papernot et al. (2018), one can use the formal framework of sub-sample and aggregate in conjunction with Propose-Test-Release (PTR) for this selection. The algorithm first splits the training data into disjoint subsets, and trains models independently on each of the splits. Using these trained models, the parameter is chosen via consensus voting with differential privacy. Notice that if the training data set is large, and there is a strong consensus, then the cost towards

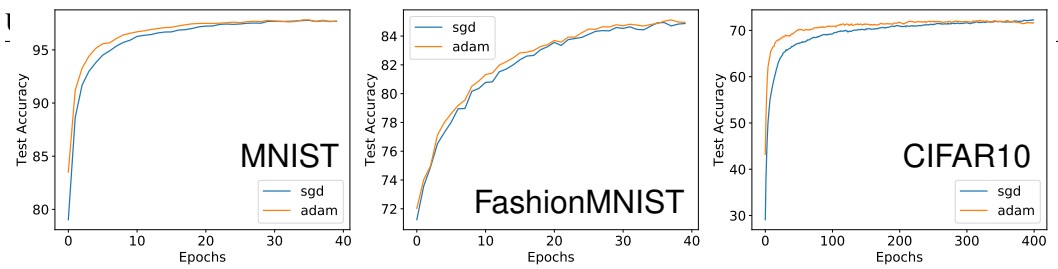

Figure 5: Learning curves for DP-SGD and DP-Adam. Early on in training, DP-Adam converges faster to an accuracy that is within 1 point of its final accuracy, however DP-SGD increases more steadily towards the end of training, thus both achieve comparable results. Given one of the datasets, the privacy budget $\varepsilon$ for both models is identical at each epoch.

privacy is very low.) The idea is that the mean and variance pairs can be obtained quickly at a modest privacy budget, but the faster convergence of the *Mean Var* initialized model both reduces the overall privacy budget needed for training, and mitigates the increased wall-clock time of DP-SGD.

We experiment with a relatively deep CIFAR10 convolutional model (see Appendix A), since Raghu et al. found the benefits of *Mean Var* initialization most pronounced for large models. We first trained a model using random initialization, and then did weight scaling by transferring that model's statistics to a new model. In this proof-of-concept, both models were trained with the same noise variance ($\sigma = 0.5$), but one could reserve a larger portion of the privacy budget for the new model.

We should note that we did not directly transfer the weight statistics between corresponding layers in the original and new models. Rather, we used the weight statistics of each of original model's early layers of the original model for two of the layers in the new model. This gives superior performance to a one-to-one transfer; we conjecture that this is because early layers have higher variance.

Figure 4 shows the results of this experiment for some early training epochs. Each run that used standard He random initialization (He et al., 2015) gave near identical results, achieving 37% accuracy at epoch 33. The *Mean Var* initialization runs showed substantially higher variance, with the best models having 7% better accuracy at epoch 33. These results are intriguing, and reminiscent of the lottery ticket hypothesis (Frankle & Carbin, 2019); they suggest a strategy of training a collection of *Mean Var* models and keeping those that show early promise.

## 5 TUNING OPTIMIZERS FOR PRIVATE LEARNING

Architectural choices presented in Section 3 control how sensitive learning is to training examples. This helps us to learn with privacy—because it eliminates the negative effects of clipping and noising large gradients. We now turn our attention to the training algorithm itself. We find that it is important to tailor algorithm and hyperparameter choices to the specificities of private learning: a batch size or learning rate that yields good results without privacy may not perform well with privacy.

### 5.1 ADAPTIVE OPTIMIZERS PROVIDE MARGINAL GAINS WHEN LEARNING WITH PRIVACY

We first explore the choice of optimizer, and in particular whether adaptive optimizers that leverage the history of iterates help convergence when learning privately. We compare learning curves for DP-SGD and the differentially private counterpart of Adam (Kingma & Ba, 2014), a canonical adaptive optimizer. A qualitative analysis of Figure 5 leads to the same conclusion for all datasets (MNIST, FashionMNIST, and CIFAR10). While DP-Adam may converge faster initially, its convergence rate eventually slows down sufficiently for DP-SGD to achieve comparable (if not higher) accuracy.

To explain the ineffectiveness of adaptive optimizers, we hypothesize that the iterates they accumulate during training are affected negatively by noise introduced to preserve privacy. Indeed, while there is enough signal from the training data included in any given batch sampled early in training, later in training most training examples have a loss of zero and do not contribute to the gradients being noised. Carrying this noise from one gradient descent step to the next to adapt learning rates therefore inadequately slows down training. To verify this, we track the estimate of the first moment in Adam on MNIST. The mean absolute value of its components converges when learning without privacy (from 0.5 after the first epoch to about 0.8 for epochs 45 through 60). Instead, it increases steadily throughout training with privacy (from 0.5 at the first epoch to above 1. after 60 epochs).

Thus, choosing an adaptive optimizer (e.g., DP-Adam) is not necessary if one is interested in achieving maximal accuracy: given a fixed privacy budget, fine-tuning the learning rate is more important

as we confirm in Section 5.2. Note that this resonates well with recent results questioning the generalization capabilities of adaptive optimizers (Wilson et al., 2017).

## 5.2 CHOOSING A (LARGE) BATCH SIZE AND LEARNING RATE

Having observed that few training examples contribute signal after the first epochs, it is natural to ask whether increasing the size of batches could improve the noise-to-signal ratio in DP learning.

To ensure a fair comparison, we fix the privacy budget $\varepsilon$ and deduce the number of epochs we can train the model for given a desired batch size. For instance, in Table 5, we compare models trained for 7 epochs on batches of $1,024$ examples to models trained for 40 epochs on batches of 256 examples. In both cases, the total privacy budget for training these models is $\varepsilon = 2.7$. We run a hyperparameter search to fine-tune the choice of learning rate for both DP-SGD and DP-Adam. We then compare the test accuracy achieved with small and large batch sizes.

| Optimizer | Batch | Epochs | Time | Non-private | | Differentially-private | |
|---|---|---|---|---|---|---|---|
| | | | | Learning Rate | Test Acc. | Learning Rate | Test Acc. |
| SGD | 256 | 40 | 240s | $1.07 \cdot 10^{-1}$ | 90.3% | $3.32 \cdot 10^{-1}$ | 86.1% |
| | 1024 | 7 | 42s | $3.68 \cdot 10^{-1}$ | 86.3% | 4.46 | 85.1% |
| Adam | 256 | 40 | 240s | $1.06 \cdot 10^{-3}$ | 90.5% | $1.32 \cdot 10^{-3}$ | 86.0% |
| | 1024 | 7 | 42s | $4.32 \cdot 10^{-3}$ | 88.7% | $7.08 \cdot 10^{-3}$ | 85.1% |

Table 5: Impact of batch size on trade-off between accuracy and privacy. The privacy budget is fixed to $\varepsilon = 2.7$ for all rows. A hyperparameter search is then conducted to find the best learning rate to train the model with or without differential privacy on FashionMNIST.

**Hyperparameters should be tuned for DP-SGD, not SGD.** We confirm that DP-Adam does not improve over DP-SGD. Yet, this experiment shows how training for a small number of epochs at a large batch size can do comparably well to a large number of epochs at a small batch size: the wall-clock time gain is important (about $5\times$) and the cost in performance is moderate—half a percentage point. This suggests that earlier theoretical analysis (Talwar et al., 2014) also holds in the non-convex setting. Furthermore, note how learning rates vary across the non-DP and DP settings.

## 6 CONCLUSIONS

Rather than first train a non-private model and later attempt to make it private, we bypass non-private training altogether and directly incorporate specificities of privacy-preserving learning in the selection of architectures, initializations, and tuning strategies. This improves substantially upon the state-of-the-art privacy/accuracy trade-offs on three benchmarks, as summarized below. Up to now, we evaluated each component (e.g., change of activation function, optimizer, etc.) individually to demonstrate its influence on private learning. Instead, here this summary table compares each approach after all hyperparameters explored in the paper have been jointly fined tuned. In particular, note how even in their own individually-best setting, tanh continues to consistently outperform ReLU with for example 98.1% test accuracy (instead of 96.6% for ReLU) on MNIST.

| Dataset | Technique | Acc. | $\varepsilon$ | $\delta$ | Assumptions |
|---|---|---|---|---|---|
| MNIST | SGD w/ tanh (not private) | 99.0% | $\infty$ | 0 | - |
| | DP-SGD w/ ReLU | 96.6% | 2.93 | $10^{-5}$ | - |
| | **DP-SGD w/ tanh (ours)** | **98.1%** | **2.93** | $10^{-5}$ | - |
| FashionMNIST | SGD w/ ReLU (not private) | 89.4% | $\infty$ | 0 | - |
| | DP-SGD w/ ReLU | 81.9% | 2.7 | $10^{-5}$ | - |
| | **DP-SGD w/ tanh (ours)** | **86.1%** | **2.7** | $10^{-5}$ | - |
| CIFAR10 | Transfer + SGD (not private) | 75% | $\infty$ | 0 | - |
| | Transfer + DP-SGD (Abadi et al.) | 67% | 2 | $10^{-5}$ | Public Data |
| | **Transfer + DP-SGD (ours)** | **72%** | **2.1** | $10^{-5}$ | Public Data |

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

## A  DEEP CONVOLUTIONAL MODEL

| Conv | 64 filters of 3x3, strides 1 |
|---|---|
| Conv | 128 filters of 3x3, strides 1 |
| Av pooling | 2x2 |
| Conv | 128 filters of 3x3, strides 1 |
| Conv | 256 filters of 3x3, strides 1 |
| Av pooling | 2x2 |
| Conv | 256 filters of 3x3, strides 1 |
| Conv | 512 filters of 3x3, strides 1 |
| Av pooling | 2x2 |
| Conv | 10 filters of 3x3, strides 1 |
| Reduce mean | 1x2 |
| Softmax | 10 units |

Table 6: All convolutional architecture for CIFAR10 model with 2,334,730 parameters.

## B  EXPERIMENTAL DETAILS

We describe hyperparameters used in Sections 3, 4, and 5 because they were omitted from the main body of the paper due to space constraints.

| | Section 3.1 | Section 3.2 |
|---|---|---|
| Batch size / microbatches | 100 | 256 |
| Epochs | 40 | 40 |
| Optimizer | SGD / DP-SGD | SGD / DP-SGD |
| Learning rate | 0.15 | 0.15 |
| Clipping norm | 1.0 | 1.0 |
| Noise multiplier | 1.1 | 1.1 |
| Architecture | Table 1 (varying $k$) | Table 1 ($k = 16$) |
| Activation function | ReLU | ReLU / tanh |

| | Section 4.1 (CIFAR100) | Section 4.1 (CIFAR10) | Section 4.2 |
|---|---|---|---|
| Batch size / microbatches | 64 | 5000 | 500 |
| Epochs | 150 | 400 | See Figure 4 |
| Optimizer | Adam | DP-SGD | DP-Adam |
| Learning rate | 0.001 | 0.8 | 0.0001 |
| Clipping norm | n/a (public) | 1.0 | 1,0 |
| Noise multiplier | n/a (public) | 15 | 0.5 |
| Architecture | Table 2 | Table 2 | Table 6 |
| Activation function | ReLU | ReLU | tanh |

| | Section 5.1 | Section 5.2 |
|---|---|---|
| Batch size / microbatches | 256 | 256 / 1024 |
| Epochs | 40 | 40 / 7 |
| Optimizer | DP-SGD / DP-Adam | SGD / DP-SGD / Adam / DP-Adam |
| Learning rate | See Table 5 for dataset specific rates | |
| Clipping norm | 1.0 | 1.0 |
| Noise multiplier | 1.1 | 1.1 |
| Architecture | Table 1 ($k = 16$) | Table 1 ($k = 16$) |
| Activation function | tanh | tanh |

