# OpenReview forum: "Making the Shoe Fit: Architectures, Initializations, and Tuning for Learning with Privacy"
_ICLR.cc/2020/Conference — Reject_

### Official Review · AnonReviewer1 · 2019-10-24
**Official Blind Review #1**

**Rating:** 6

**Review:**

The paper methodically analyses the settings and choices used when training neural networks (specifically CNNs) via the DP-SGD algorithm and suggests changes to the standard procedures that empirically lead to higher accuracies despite the added noise. The main statement of the paper is quite simple: optimize hyperparameters for the model that you're training (DP-SGD) rather than the model it is inspired by. Yet, the findings an recommendations may be useful for practitioners.

Nevertheless, to be more practically relevant the paper needs some modifications:

The example models used in demonstrations are quite small (3 hidden layers 26,000 parameters, when, for example, a standard segmentation CNN model U-net can typically have 26,000,000, AlexNet has about 60,000,000 and so on). The results would be much more convincing if these or other models widely used in practice were used as running examples.

In Figure 1 the multitude of point on the plot makes it unclear whether they represent result variability per number of filters or simply reflect variability as the number of filters grows. If it is the latter, it seems appropriate to perform a cross validation analysis and report standard deviations. Especially in the MNIST plot the values for SGD and DP-SGD are so close that they may in fact be statistically indistinguishable. Hard to tell by looking at a point estimate. The same request holds for Figure 2, where the difference may be immaterial, but as the figure currently stands it is unclear.

Section 3.2 reports some numbers for test accuracy but the uncertainty of these numbers with respect to the test set changes (cross validation) is not reported and the numbers are quite close to each other. Furthermore, the dataset is not described and it is unclear what was the size of the training and the test sets.

Conclusions relative Adam vs SGD seem to repeat what's already known or been discussed about these methods outside of the DP topic. May be worth highlighting that when one knows how to set learning rates for SGD (may be via learning rate scheduler, not discussed in the paper but practically relevant) then SGD may be as good or slightly better than Adam. However note, adaptive optimizers are often preferred for their ease of use as no tweaking and searching for an optimal learning rate is required. Would not this problem be detrimental for SGD optimization affecting the privacy budget?

Please add wall-clock time column to Table 5 to support the statement about 4 times gain.

I think it's more accurate to change " This confirms that earlier theoreticalanalysis (Talwar et al., 2014) also holds in the non-convex setting." to "This suggests that earlier theoreticalanalysis (Talwar et al., 2014) also holds in the non-convex setting."



**Experience Assessment:**

I have published in this field for several years.

**Review Assessment: Checking Correctness Of Derivations And Theory:**

N/A

**Review Assessment: Checking Correctness Of Experiments:**

I carefully checked the experiments.

**Review Assessment: Thoroughness In Paper Reading:**

I read the paper thoroughly.

---

> ### Author Response · Authors · 2019-11-15
> **Thank you for the review.**
>
> > The example models used in demonstrations are quite small (3 hidden layers 26,000 parameters, when, for example, a standard segmentation CNN model U-net can typically have 26,000,000, AlexNet has about 60,000,000 and so on). The results would be much more convincing if these or other models widely used in practice were used as running examples.
>
> We agree that larger datasets and larger models would be more convincing (e.g., ImageNet with AlexNet or a more modern architecture). However, research into differentially-private ML (DPML) has simply not progressed to be able to offer strong privacy/utility tradeoffs for such models and tasks. A quick literature survey (or Google Scholar search) shows that DPML work often considers only simple tasks (e.g., logistic regression on the UCI Adult dataset), even in 2019. For DPML work with strong privacy guarantees and high utility, the most challenging datasets considered are still those of MNIST, FashionMNIST, and CIFAR-10 (e.g., see the recent survey in Jayaraman & Evans, https://arxiv.org/pdf/1902.08874.pdf). The DPML research community has been focused on improving the accuracy of those models, without sacrificing the DP privacy guarantees (i.e., without increasing the DP epsilon), as we do in our work, reaching a new state-of-the-art.  (This said, we strongly agree that DPML research should move onto more complex tasks; we feel that the results in our current paper are a good step in that direction.)
>
> We chose relatively simple and small models for our MNIST experiments because no further  complexity or capacity was needed to achieve good accuracy without privacy. Because DPSGD training becomes increasingly more challenging with increased dimensionality (see Section 3.1), a large number of superfluous model parameters can only hinder DPML training to high accuracy. Choosing larger models would have handicapped our experiments, unnecessarily.
>
> > In Figure 1 the multitude of point on the plot makes it unclear whether they represent result variability per number of filters or simply reflect variability as the number of filters grows. If it is the latter, it seems appropriate to perform a cross validation analysis and report standard deviations. Especially in the MNIST plot the values for SGD and DP-SGD are so close that they may in fact be statistically indistinguishable. Hard to tell by looking at a point estimate. The same request holds for Figure 2, where the difference may be immaterial, but as the figure currently stands it is unclear.
>
> Originally, Figure 1 plotted the outcome of a fine-tuning strategy optimizing the number of filters to maximize accuracy, which explained the multitude of points in the regions with the largest accuracy (high number of filters for SGD and low number of filters for DP-SGD). We updated Figure 1 to instead report the mean accuracy for each number of filters along with the standard deviation. This updated figure clearly indicates that there is an inflection point for DP-SGD, which does not exist for SGD, on both datasets.
>
> > Section 3.2 reports some numbers for test accuracy but the uncertainty of these numbers with respect to the test set changes (cross validation) is not reported and the numbers are quite close to each other. Furthermore, the dataset is not described and it is unclear what was the size of the training and the test sets.
>
> We’ve added standard deviations across 10 runs for the results presented in Figure 2 within Section 3.2, demonstrating that the improvement is meaningful. To put the increase in accuracy into context, we note that on MNIST at comparable privacy guarantees, the state-of-the-art accuracy went up from 95% in 2016 (with PCA dimensionality reduction) to 96.6% in late 2018 (without any dimensionality reduction), which our results then improved to 98.1% (again without any dimensionality reduction). We’ve improved the writing to more clearly state that MNIST (Figure 2 - left) and FashionMNIST (Figure 2 - right) refer to their standard learning tasks, and their datasets of 60,000 training examples and 10,000 test examples, as is standard.

---

> > ### Author Response · Authors · 2019-11-15
> > **Second part of comment**
> >
> > > Conclusions relative Adam vs SGD seem to repeat what's already known or been discussed about these methods outside of the DP topic. May be worth highlighting that when one knows how to set learning rates for SGD (may be via learning rate scheduler, not discussed in the paper but practically relevant) then SGD may be as good or slightly better than Adam. However note, adaptive optimizers are often preferred for their ease of use as no tweaking and searching for an optimal learning rate is required. Would not this problem be detrimental for SGD optimization affecting the privacy budget?
> >
> > The reviewer correctly points out that our Adam and SGD observations confirm what has been discussed before outside of differential privacy (we had tried to make this connection explicit through the reference to Wilson et al. at the end of Section 5.1 but are open to suggestions on how to make this more clear). Those observations deserved to be revisited in the context of DPML, where noise and clipping are confounding factors. For DPML, we wanted to raise awareness to how---particularly in later epochs---SGD may outperform Adam due to the accumulation of noise injected to preserve privacy (see Figure 5-right), and how some privacy budget could be allocated to fine-tuning the learning rate with privacy [X].
> >
> > [X] Liu, Jingcheng, and Kunal Talwar. "Private selection from private candidates." Proceedings of the 51st Annual ACM SIGACT Symposium on Theory of Computing. ACM, 2019.
> >
> > > Please add wall-clock time column to Table 5 to support the statement about 4 times gain.
> >
> > We’ve added wall-clock time to Table 5 to support this statement.
> >
> > I think it's more accurate to change " This confirms that earlier theoretical analysis (Talwar et al., 2014) also holds in the non-convex setting." to "This suggests that earlier theoreticalanalysis (Talwar et al., 2014) also holds in the non-convex setting."
> >
> > Thank you for the suggestion, we fixed the language around the citation to Talwar et al., 2014.

---

### Official Review · AnonReviewer2 · 2019-10-24
**Official Blind Review #2**

**Rating:** 3

**Review:**

Overall, this work empirically evaluates different techniques used in privacy learning and suggest useful methods to stabilize or improve performance.

Detail comments:

Strength:
Despite the progress of privacy-preserving learning in theory, there are few works providing learning details for better training. Especially, considering the instability in perturbation-based private algorithms, e.g., most DP ones, the work could be valuable in the sense of practice.

Weakness:
As far as empirical research, the compared techniques are too few. What if we use those less popular techniques, for example, RMSprop optimization method?

The model capacity of neural networks, especially deep networks, has some non-trivial relation to the number of filters or the number parameters. It is important to quantify such relation. A good reference might be [A]. Briefly, the generalization performance may not be monotonic against the number of parameters.

The baselines are not enough. Of course, Abadi et al.’s work is outstanding in handling the privacy learning of deep networks. It has been further developed by the following researchers. For example, [B] and [C]. Does the conclusion still hold for these algorithms?

[A] Neyshabur, B., Bhojanapalli, S., Mcallester, D., & Srebro, N. (2017). Exploring Generalization in Deep Learning. In I. Guyon, U. V. Luxburg, S. Bengio, H. Wallach, R. Fergus, S. Vishwanathan, & R. Garnett (Eds.), Advances in Neural Information Processing Systems 30 (pp. 5947–5956).
[B] Yu, L., Liu, L., Pu, C., Gursoy, M. E., & Truex, S. (2019). Differentially Private Model Publishing for Deep Learning. Proceedings of 40th IEEE Symposium on Security and Privacy.
[C] Phan, N., Vu, M. N., Liu, Y., Jin, R., Dou, D., Wu, X., & Thai, M. T. (2019). Heterogeneous Gaussian Mechanism: Preserving Differential Privacy in Deep Learning with Provable Robustness. Proceedings of the Twenty-Eighth International Joint Conference on Artificial

**Experience Assessment:**

I have published one or two papers in this area.

**Review Assessment: Checking Correctness Of Derivations And Theory:**

I did not assess the derivations or theory.

**Review Assessment: Checking Correctness Of Experiments:**

I carefully checked the experiments.

**Review Assessment: Thoroughness In Paper Reading:**

I made a quick assessment of this paper.

---

> ### Author Response · Authors · 2019-11-15
> **Thank you for the review.**
>
> > As far as empirical research, the compared techniques are too few. What if we use those less popular techniques, for example, RMSprop optimization method?
>
> We believe the techniques considered are sufficiently broad in scope because they enable us to improve the state-of-the-art significantly. In particular, we focused on SGD and Adam because they are the most popular optimizers. While we did not report results for other optimizers, our preliminary experiments show similar conclusions for other adaptive optimizers such as SGD with momentum or RMSprop. Recall that training with differential privacy is slow (in terms of wall-clock training time) because one needs to compute per-example gradients of the loss rather than gradients of the average loss across a batch of examples. This limited our ability to repeat all of our experiments with other optimizers within the scope of the rebuttal process.
>
> > The model capacity of neural networks, especially deep networks, has some non-trivial relation to the number of filters or the number parameters. It is important to quantify such relation. A good reference might be [A]. Briefly, the generalization performance may not be monotonic against the number of parameters.
>
> Thank you for the pointer, we added a reference to [A] in our revised manuscript. As shown in Figure 4 of [A], the number of parameters is a good proxy for the model’s capacity in our setting. However, we acknowledge that the relationship between generalization performance and the number of parameters is not always monotonic. In fact, we believe that a future study of how different measures of capacity can inform the design of model architectures for private learning would be fruitful. We’ve updated the paper to reflect insights provided by your comment.
>
> [A] Neyshabur, B., Bhojanapalli, S., Mcallester, D., & Srebro, N. (2017). Exploring Generalization in Deep Learning. In I. Guyon, U. V. Luxburg, S. Bengio, H. Wallach, R. Fergus, S. Vishwanathan, & R. Garnett (Eds.), Advances in Neural Information Processing Systems 30 (pp. 5947–5956).

---

> > ### Author Response · Authors · 2019-11-15
> > **Second part of comment.**
> >
> > > The baselines are not enough. Of course, Abadi et al.’s work is outstanding in handling the privacy learning of deep networks. It has been further developed by the following researchers. For example, [B] and [C]. Does the conclusion still hold for these algorithms?
> >
> > We regret not having provided provided more up-to-date baselines than those in Abadi et al., and thank the reviewer for pointing this out.
> >
> > For MNIST, the previous SotA in terms of privacy and utility is---as far as we know---that shown in the TensorFlow Privacy GitHub repo: 95% accuracy at eps 1.19;  96.6% at eps 3.01, and  97% accuracy at epsilon 7.10 (see https://github.com/tensorflow/privacy/tree/master/tutorials)
> >
> > For CIFAR10, the previous SotA is harder to nail down, because of the number of incomparable assumptions and privacy definitions in previous work. However, for the setup that we use, and was also used in Abadi el at. (CIFAR-10 training with transfer learning from CIFAR-100), we know of no result that has improved on the initial results of Abadi et al.
> >
> > Both [B] and [C] are based on standard DP-SGD; therefore, all of our techniques should be directly applicable. With regard to baselines, the MNIST accuracy is less than 94% in [B] and less than 60% in [C, Figure 5] and the CIFAR10 accuracy is less than 45% in both papers, for all privacy levels that the papers investigate (which seem along the same lines as in our work).
> >
> > Below are further, recent papers that we can discuss in the camera ready, if the reviewers feel this is appropriate.
> >
> > As another baseline, we considered discussing [D], as it reports very high accuracy and privacy for MNIST. However, we decided against it, because over the last year-and-a-half we have been unable to replicate the paper’s results. Even when using code provided by the authors, we have only trained models with privacy/accuracy tradeoffs much worse than those reported elsewhere (e.g., in TensorFlow Privacy). As part of writing this ICLR response, we contacted the authors again, and they told us that they are now themselves unable to reproduce their results.
> >
> > One of several recent papers based on distributed/federated means of learning is [E], which reports MNIST accuracy of 97% and CIFAR10 accuracy of 94% for a 3-layer CNN, both at epsilon 4.65. However, [E] does not fully explain how those results can have come about or, in particular, how the leader/follower architecture may have strengthened the per-example signal and changed the meaning of epsilon upper bounds. Distributed learning can radically change privacy definitions and transmit information in unexpected ways (e.g., see [G]); therefore, we decided against comparing against [E] until we have a better understanding of the work.
> >
> > Finally, [F] describes another distributed mechanism for learning with privacy that reports excellent privacy/utility tradeoffs for both CIFAR10 and MNIST (although the MNIST results are worse than in our work). However, the results in [F] do not seem to be using the same definition of DP epsilons as is standard, and as are used in our work. This can be seen in Figure 10, where model accuracy remains unchanged despite epsilon increasing from 1 to 10.  This can also be seen in Table II, where there is a multi-percentage gap between training accuracy and test accuracy---even though a defining characteristic of DPML with significant epsilons is that it should eliminate any such gap, and does do so in practice.
> >
> > [B] Yu et al. Differentially Private Model Publishing for Deep Learning. IEEE S&P, 2019 (version at https://arxiv.org/abs/1904.02200)
> >
> > [C] Phan et al. Heterogeneous Gaussian Mechanism: Preserving Differential Privacy in Deep Learning with Provable Robustness. Joint Conf on AI, 2019 https://arxiv.org/abs/1906.01444
> >
> > [D] Li et al. On Connecting Stochastic Gradient MCMC and Differential Privacy, PMLR 2019 (http://proceedings.mlr.press/v89/li19a.html; Arxiv version is incorrect, according to authors.)
> >
> > [E] Cheng et al. LEASGD: an Efficient and Privacy-Preserving Decentralized Algorithm for Distributed Learning, PPML 2018, at https://arxiv.org/pdf/1811.11124.pdf
> >
> > [F] Arachchige et al. Local Differential Privacy for Deep Learning, Internet of Things Journal, 2019, at https://arxiv.org/pdf/1908.02997.pdf
> >
> > [G] Melis et al. Exploiting Unintended Feature Leakage in Collaborative Learning, IEEE S&P 2019, https://arxiv.org/pdf/1805.04049.pdf

---

### Official Review · AnonReviewer4 · 2019-11-04
**Official Blind Review #4**

**Rating:** 6

**Review:**

This paper presents experimental evidence that learning with privacy requires approaches that are not identical to those used when learning without privacy. These approaches include re-considering different model choices (i.e., its structure and activation functions), its initialization, and its optimization procedure. With these changes, they show that it is possible to obtain state-of-the-art results for some canonical learning tasks.

Strengths:
This paper questions nearly every component in the training pipeline, including choices about the model structure, initialization strategies, and optimization procedures. For each component, they show that judiciously choosing the components (which go against the standard choices in the non-private learning setting) enables training higher-utility models than in previous works without sacrificing privacy. Moreover, in addition to the experimental evidence alone, most of the components considered in the paper were accompanied by reasonable justification/hypotheses for why the choices enable such improvements.
This paper helps push differentially private learning to a more practically-useful realm. First, the suggested changes here are easy for a practitioner to understand and easy to implement. With only these simple changes, the concrete results then show that it is possible to achieve utility close to the analogous non-private model while still maintaining reasonable utility (\epsilon less than 3 with \delta of 10^-5).


Weaknesses:
My major concern with this paper lies in the experimental methodology. Specifically, most experiments are based on varying a single component while leaving all other components the same. While this is certainly the scientifically-valid way to demonstrate the component’s influence on the entire system given the other fixed components, it doesn’t convincingly demonstrate that the component has this influence across all (or at least most) reasonable configurations of the other components.
This can be made concrete using many experiments in the paper, but let’s take the activation functions experiment of 3.2 as an example. Here, it is shown that after fixing the privacy guarantee, model structure, training procedure, and hyperparameters -- the tanh activation performs better than the ReLU activation. However, suppose instead that we fix all of these components except the hyperparameters; it may then be the case that the ReLU activation is capable of outperforming the tanh activation when its hyperparameters are chosen carefully. In other words, to validly compare the two activations and reach a convincing conclusion, they should be compared against each other in their own individually-best settings (e.g., the results induced by the optimal hyperparameters for ReLU versus the results induced by the optimal hyperparameters for tanh).
This is similar to the problem addressed in Avent et al.’s “Automatic Discovery of Privacy–Utility Pareto Fronts” paper (https://arxiv.org/abs/1905.10862).
The specific technical details on some experiments were either difficult to find or were lacking. Given that this is fundamentally an experimental paper, having these details clearly listed somewhere for reference is important, even if relegated to an Appendix. Although this applies more broadly to most of the experiments, we can use Section 3 as an example again: the details on the experiment in 3.1 were found in the caption of Figure 1, whereas I would have expected them either in the main body or clearly listed in their own table; the details on the experiment in 3.2 specify that everything is identical between the tests of the two activation functions, however it is never specified exactly what is being altered (and by how much) to vary the \epsilon value.
4.2 Initialization by Weight Scaling proposes that judiciously scaling initial weights can improve model privacy/utility. This scaling is done by “transfer from one differentially private model to another”, where “DP-SGD can be applied to train a model with high utility, but less than ideal privacy” and then extracting the relevant information from there in order to initialize a new differentially private model that will be trained with strong privacy guarantees. It is claimed that “this extraction can be done in a differentially-private manner, e.g., as in Papernot et al. (2018), although the privacy
risk of summary statistics that drive random initialization should be vanishing”. It is unclear to me how this extraction of summary statistics should be done in such a way that doesn’t consume a significant portion of the privacy budget. If there is such a way, it should be clearly stated and its effect on the privacy budget should be explicitly incorporated into this paper’s results.
Minor: The statement that “Such accuracy loss may sometimes be inevitable” on page 1 should include a reference; e.g., Feldman’s “Does Learning Require Memorization? A Short Tale about a Long Tail” paper (https://arxiv.org/abs/1906.05271).


Overall, this work provides good practical guidance to practitioners and researchers who wish to do differentially private machine learning. However, given the lack of theoretical novelty, the experimental methodology needs to be improved in order to significantly strengthen the results (assuming they continue to hold).


----------------------------------------------------

Update: Due to the authors' writing clarifications and experimental additions, in conjunction with the concrete and realistically-applicable insights from the paper, I've modified my rating to a Weak Accept.

**Experience Assessment:**

I have published one or two papers in this area.

**Review Assessment: Checking Correctness Of Derivations And Theory:**

N/A

**Review Assessment: Checking Correctness Of Experiments:**

I carefully checked the experiments.

**Review Assessment: Thoroughness In Paper Reading:**

I read the paper at least twice and used my best judgement in assessing the paper.

---

> ### Author Response · Authors · 2019-11-15
> **Thank you for the review.**
>
> > My major concern with this paper lies in the experimental methodology. Specifically, most experiments are based on varying a single component while leaving all other components the same. While this is certainly the scientifically-valid way to demonstrate the component’s influence on the entire system given the other fixed components, it doesn’t convincingly demonstrate that the component has this influence across all (or at least most) reasonable configurations of the other components.
> This can be made concrete using many experiments in the paper, but let’s take the activation functions experiment of 3.2 as an example. Here, it is shown that after fixing the privacy guarantee, model structure, training procedure, and hyperparameters -- the tanh activation performs better than the ReLU activation. However, suppose instead that we fix all of these components except the hyperparameters; it may then be the case that the ReLU activation is capable of outperforming the tanh activation when its hyperparameters are chosen carefully. In other words, to validly compare the two activations and reach a convincing conclusion, they should be compared against each other in their own individually-best settings (e.g., the results induced by the optimal hyperparameters for ReLU versus the results induced by the optimal hyperparameters for tanh). This is similar to the problem addressed in Avent et al.’s “Automatic Discovery of Privacy–Utility Pareto Fronts” paper (https://arxiv.org/abs/1905.10862).
>
> We thank the reviewer for their careful review and suggestions on methodology and exposition. Due to poor presentation of our experimental methodology, it was unclear in our original submission that we had simultaneously finetuned all components, explored individually in the body of the paper, to produce the summary table included in the conclusion. We’ve updated the language in the conclusion to explain how this summary table for instance presents experimental results that compare ReLU with tanh in their own individually-best setting, by first fixing the activation function and then fine-tuning all other hyperparameters (model structure, training procedure, and hyperparameters). It shows that tanh consistently outperforms ReLU: e.g., with 98.1% test accuracy instead of 96.6% test accuracy on MNIST for the same privacy guarantee, even in their own individually-best settings. We also added a citation to Avent et al. in the main body of the paper.
>
> > The specific technical details on some experiments were either difficult to find or were lacking. Given that this is fundamentally an experimental paper, having these details clearly listed somewhere for reference is important, even if relegated to an Appendix. Although this applies more broadly to most of the experiments, we can use Section 3 as an example again: the details on the experiment in 3.1 were found in the caption of Figure 1, whereas I would have expected them either in the main body or clearly listed in their own table; the details on the experiment in 3.2 specify that everything is identical between the tests of the two activation functions, however it is never specified exactly what is being altered (and by how much) to vary the \epsilon value.
>
> We added missing details of the experimental setup in an Appendix.

---

> > ### Author Response · Authors · 2019-11-15
> > **Second part of the comment.**
> >
> > > 4.2 Initialization by Weight Scaling proposes that judiciously scaling initial weights can improve model privacy/utility. This scaling is done by “transfer from one differentially private model to another”, where “DP-SGD can be applied to train a model with high utility, but less than ideal privacy” and then extracting the relevant information from there in order to initialize a new differentially private model that will be trained with strong privacy guarantees. It is claimed that “this extraction can be done in a differentially-private manner, e.g., as in Papernot et al. (2018), although the privacy
> > risk of summary statistics that drive random initialization should be vanishing”. It is unclear to me how this extraction of summary statistics should be done in such a way that doesn’t consume a significant portion of the privacy budget. If there is such a way, it should be clearly stated and its effect on the privacy budget should be explicitly incorporated into this paper’s results.
> >
> > We’ve clarified in the paper how extracting these summary statistics should be done without consuming a significant portion of the privacy budget. Following [Y, Z], one can use the formal framework of sub-sample and aggregate in conjunction with Propose-Test-Release (PTR) for this selection. Given privacy parameters (\epsilon, \delta), the algorithm first splits the training data into disjoint subsets, and trains models independently on each of the splits. Using these trained models, the parameter is chosen via consensus voting with differential privacy. To ensure privacy, a consensus of majority + (1/\epsilon) * \log(1/\delta) is needed. Notice that if the training data set is large, and there is a strong consensus, then the cost towards privacy is very low.
> >
> > [Y] Bassily, Raef, Om Thakkar, and Abhradeep Guha Thakurta. "Model-agnostic private learning." Advances in Neural Information Processing Systems. 2018.
> >
> > [Z] Papernot, Nicolas, et al. "Scalable private learning with pate." ICLR (2018).
> >
> > > Minor: The statement that “Such accuracy loss may sometimes be inevitable” on page 1 should include a reference; e.g., Feldman’s “Does Learning Require Memorization? A Short Tale about a Long Tail” paper (https://arxiv.org/abs/1906.05271).
> >
> > Feldman’s paper provides excellent support for our statement.  We added a citation to it, as well as to https://arxiv.org/abs/1905.12101 which makes related empirical observations.

---

> > > ### Author Response · Authors · 2019-11-15
> > > **New weight-scaling experiments**
> > >
> > > Following the reviewer's advice to make it easy to understand the worst-case potential privacy implications of weight scaling, we performed new weight-scaling experiments, now shown in Figure 4.  These experiments all transfer the mean and variance from seed models previously-trained with differential privacy using DP-SGD with the same clipping and noise settings.  The experiments exhibit the same type of improvements as shown in the original submission where the transfer was performed from a non-private model.
> > >
> > > By their construction, the worst-case privacy loss is at most doubled in these experiments. Since we are post-processing an already-DP model, this transfer does not require any complexities, such as training multiple models, doing PATE-style PTR, etc., although we still discuss this as a means of extracting statistics. (Of course, since only six floating-point values---three mean and variance pairs---are extracted from a model trained with privacy, and used to bias a random initialization process, the real-world privacy loss is likely to increase by less than a factor of two.)

---

### Decision · Program_Chairs · 2019-12-19

**Decision:**

Reject

**Comment:**

This paper presents experimental evidence that learning with privacy requires optimization of the model settings (architectures and initializations) that are not identical to those used when learning without privacy. While acknowledging potential usefulness of this work for practitioners, the reviewers expressed several important concerns such as (1) lack of SOTA baseline comparisons, (2) lack of clarity of the empirical evaluation protocols, (3) large models (that are widely used in practice) have not been studied in the paper, (4) low technical novelty. The authors have successfully addressed some of the concerns regarding (1) and (2). However (3) and (4) make it difficult to assess the benefits of the proposed approach for the community and were viewed by AC as critical issues. We hope the detailed reviews are useful for improving and revising the paper.